# Differential Gene Expression of Checkpoint Markers and Cancer Markers in Mouse Models of Spontaneous Chronic Colitis

**DOI:** 10.3390/cancers15194793

**Published:** 2023-09-29

**Authors:** Ramya Ephraim, Sarah Fraser, Jeannie Devereaux, Rhian Stavely, Jack Feehan, Rajaraman Eri, Kulmira Nurgali, Vasso Apostolopoulos

**Affiliations:** 1Institute for Health and Sport, Victoria University, Melbourne, VIC 3030, Australia; ramya.ephraim@live.vu.edu.au (R.E.); sarah.fraser@vu.edu.au (S.F.); jeannie.devereaux@live.vu.edu.au (J.D.); jack.feehan@vu.edu.au (J.F.); kulmira.nurgali@vu.edu.au (K.N.); 2Pediatric Surgery Research Laboratories, Department of Pediatric Surgery, Massachusetts General Hospital, Harvard Medical School, Boston, MA 02114, USA; rstavely@mgh.harvard.edu; 3Immunology Program, Australian Institute of Musculoskeletal Science (AIMSS), Melbourne, VIC 3021, Australia; 4STEM/School of Science, RMIT University, Melbourne, VIC 3001, Australia; rajaraman.eri@rmit.edu.au; 5Department of Medicine Western Health, Faculty of Medicine, Dentistry and Health Sciences, The University of Melbourne, Melbourne, VIC 3010, Australia; 6Regenerative Medicine and Stem Cells Program, Australian Institute of Musculoskeletal Science (AIMSS), Melbourne, VIC 3021, Australia

**Keywords:** mucin, Muc2, inflammation, inflammatory bowel disease, colorectal cancer, *Winnie* mouse, *Muc2* missense mutation, chronic colitis

## Abstract

**Simple Summary:**

Chronic inflammation is a key driver of oncogenesis, and inflammatory bowel disease is strongly associated with the development of cancer. In this study, the *Winnie* mouse model of inflammatory bowel disease is used to show that the severity of inflammation leads to the expression of a wide range of cancer genes. This study provides important insights into the genetic basis for malignancy in inflammatory bowel disease, as well as identifying markers that could be used to screen for the development of cancer in patients.

**Abstract:**

The presence of checkpoint markers in cancer cells aids in immune escape. The identification of checkpoint markers and early cancer markers is of utmost importance to gain clarity regarding the relationship between colitis and progressive inflammation leading to cancer. Herein, the gene expression levels of checkpoint makers, cancer-related pathways, and cancer genes in colon tissues of mouse models of chronic colitis (*Winnie* and *Winnie-Prolapse* mice) using next-generation sequencing are determined. *Winnie* mice are a result of a *Muc2* missense mutation. The identification of such genes and their subsequent expression and role at the protein level would enable novel markers for the early diagnosis of cancer in IBD patients. The differentially expressed genes in the colonic transcriptome were analysed based on the Kyoto Encyclopedia of Genes and Genomes pathway. The expression of several oncogenes is associated with the severity of IBD, with *Winnie-Prolapse* mice expressing a large number of key genes associated with development of cancer. This research presents a number of new targets to evaluate for the development of biomarkers and therapeutics.

## 1. Introduction

Immunotherapy refers to the use of immune cells (antibody, T cell, or dendritic cell therapies) or their activation in vivo to manage, treat, or prevent disease. Immunotherapy includes cellular therapies, monoclonal antibody therapies, immune checkpoint inhibitors, non-specific immunotherapies, oncolytic virus therapies, T-cell therapies, immune modulators, dendritic cell therapies, and cancer vaccines [1,2,3,4,5,6]. Non-specific immunotherapy, also referred as non-specific immunomodulating agents, uses cytokines (interferon and interleukins) and adjuvants to stimulate immune cells to lyse cancer cells [7]. Immune checkpoints are expressed by cancer cells which bind to their ligands on T cells which act as a switch-off signal for T cells from attacking cancer cells. As a result, cancer cells evade T-cell responses and host immunity through the negative regulation of the immune system, such as PD-1, PD-L1, PD-L2, CTLA-4, TIM3, 2B4, the B and T lymphocyte attenuator (BTLA), and LAG3 [8,9,10,11,12,13,14,15,16,17]. Treatments that target these specific pathways to block or inhibit their actions are referred to as immune checkpoint blockade therapies [18,19,20], and several have been approved by the US Food and Drug Administration [21]. Combination immunotherapy based on immune checkpoint blockade therapies has become a first-line treatment preference for several cancers, including hepatocellular carcinoma, renal cell carcinoma, lung cancer, cervical cancer, and gastric cancer. Immune checkpoint blockade therapies in combination regimens have improved objective response rates, disease control rates, progression-free survival, and overall survival time in patients with advanced cancer compared to those receiving immune checkpoint blockade monotherapy [22,23].

Inflammatory bowel disease (IBD), namely ulcerative colitis and Crohn’s disease [24,25], is highly prevalent in developed countries. Crohn’s disease can affect any part of the gastrointestinal tract, whereas ulcerative colitis primarily affects the large colon [26]. IBD is an immune-mediated inflammatory disease involving complex interplay between host genes and environmental influences [27]. Inflammatory- and immune-related pathways/processes are highly upregulated as well as certain cancer-related genes, with those of the neurological pathways being highly downregulated in animal models of ulcerative colitis. The search for new therapeutic strategies that target immunological pathways has become of great interest, such as the IL-12/IL-23 axis, the IL-6 pathway, or Janus Kinase (JAK) inhibitors [28]. These modulate anti-inflammatory signalling pathways such as transforming growth factor-1 (TGF-1) [29], leukocyte adhesion, and migration into inflamed intestinal mucosa [30] by blocking different subunits of α4β7 integrins or other adhesion molecules [31]. Infliximab, an anti-tumour necrosis factor alpha (TNF-α) monoclonal antibody, is currently used as a first-line treatment for patients with severe IBD [32]; this results in adequate mucosal healing. However, such treatment increases susceptibility to infections and cancer [33]. Therefore, new improved and more specific immune targets are required for IBD.

Colorectal cancer (CRC) affects the caecum, colon, and rectum [34,35] and is the third most common malignancy worldwide and the second most common cause of cancer-related deaths, with more than 1 million new cases each year [36]. Colitis-associated cancer (CAC) is a subset of CRC that can develop with long-standing IBD [37,38,39,40,41,42,43]. The role of inflammation as a risk factor in CRC has been highly studied, both in CAC and in sporadic tumours [44,45,46,47]. Most patients receive chemotherapy before or after surgery; however, 80–90% of patients experience diarrhoea, constipation, oral mucositis, nausea, and vomiting. As a result, patients develop malnutrition and dehydration, leading to rapid weight loss (cachexia) [48,49]. Currently, pembrolizumab (Keytruda) and nivolumab (Opdivo) are monoclonal antibody treatments that target PD-1, and Ipilimumab (Yervoy) is a monoclonal antibody that blocks CTLA-4 [50]. Ipilimumab can be used in combination with nivolumab (Opdivo) but not used alone for CRC [51]. Novel testing methods and predictive markers are critical for enabling earlier detection and integrating rapid therapy to reduce CAC mortality and morbidity [52]. In addition, the effect of recent immunosuppressive IBD therapies on inflammation and carcinogenesis should be evaluated [53].

Although there are several mouse models of IBD, the molecular mechanisms of IBD and the early events by which inflammation promotes cancer are still not clear [54,55]. The *Winnie* mouse is a model of spontaneous chronic inflammation with immune activation similar to that seen in human IBD. *Winnie* mice have a missense mutation in the *Muc2* gene, which results in the aberrant biosynthesis of mucin, decreased mucin storage in goblet cells, and the formation of non-glycosylated *Muc2* precursor within goblet cells [56]. The *Muc2* gene expression is reduced or absent in patients with Crohn’s disease, whereas *Muc2* production and secretion are decreased in active ulcerative colitis, resulting in a thin mucus layer and increased intestinal absorption [57]. The commonalities between *Winnie* mice and patients with ulcerative colitis include changes in the morphology and motility associated with the colon, transit time, as well as microbial and metabolomic profiles [58,59]. Determining the gene expression levels of checkpoint markers like *Pd-1*, *Pd-L1*, *Pd-L2*, *Ido*, *Siglece* (ortholog of *SIGLEC* 9 in human), *Ctla4*, and *Tim3* (*Havcr2*) in the *Winnie* mouse model, as well as in severely inflamed *Winnie* mice (*Winnie-Prolapse* mice), will provide insights into disease initiation and progression [60]. In addition, analysing the genes expressed in these mice using next-generation sequencing may lead to the identification of new targets for IBD as well as early cancer markers in relation to progression from inflammation to cancer. Further, understanding the role of B7 proteins family of ligands (B7-H3/Cd276, B7-H4/Vtcn1, B7-H5/Vista, B7-H6/Ncr3lg1, and B7-H7/Hhla2) may give rise to novel checkpoint inhibitor therapies for IBD and CRC.

Overall, this project aims to deepen our understanding of the immune-related processes in IBD and CRC, emphasising the significance of pinpointing genes for early IBD and CRC detection. The identified genes and pathways will pave the way for pioneering innovative immunotherapies that improve patient outcomes while reducing the drawbacks associated with current treatments.

## 2. Materials and Methods

### 2.1. Mice and Tissues

Three groups of mice were used in this study: *Winnie* mice with inflammatory colitis (13–14 weeks, n = 3 male and n = 4 female; total n = 7), *Winnie* mice with rectal prolapse and advanced inflammation (*Winnie-Prolapse*) (18–23 weeks, n = 4 males and n = 3 females; total n = 7), and control C57BL/6 mice (14 weeks, n = 4 males and n = 4 females). *Winnie* and *Winnie-Prolapse* mice of moderate and severe chronic colitis are 2 distinct mouse models based on autonomic and neuroimaging signalling with inflammation, as previously described [60]. The *Winnie* mice, *Winnie-Prolapse* mice, and control C57BL/6 mice were obtained from Victoria University animal colony, Werribee Animal Facility (Melbourne, Australia). All mice were kept in a well-ventilated room at 22 °C with 12 h light and dark cycles and ad libitum food and water access. The Victoria University Animal Experimentation Ethics Committee approved all animal experiments, which followed the guidelines of the Australian Code of Practice for the Care and Use of Animals for Scientific Purposes (AEC18-016). All mouse faecal samples were collected, homogenised, and tested for lipocalin-2 (Lcn-2) levels to confirm intestinal inflammation using the NGAL mouse ELISA kit (Abcam, Australia). The *Winnie* mice with the most inflammation (n = 7–8; males and females) were chosen for RNA extraction from colon tissues.

### 2.2. RNA Extraction

Total RNA was extracted using TRIzol^TM^ (Invitrogen, Australia) and purified using the RNeasy^®^ Mini kit (Qiagen, Hilden, Germany), which included DNase treatment. TissueLyser EXPAND was used to homogenise colon tissues, and the total RNA samples’ integrity was determined using the Agilent 2100 Bioanalyser’s RNA 6000 Nano chip (Agilent Biotechnologies). An RNA Integrity Number of >8.5 was used for RNAseq analysis. The purity of the RNA was determined using a Denovix DS-11 spectrophotometer to measure the 260/280 and 260/230 ratios, and the concentration of individual RNA samples was quantified using an Invitrogen Qubit RNA BR assay (Gene Target Solutions, Sydney, Australia).

### 2.3. Library Preparation and RNA Sequencing

The RNA-Seq library was constructed with more than 100 ng of total RNA, including RNA quality control (CE integrity analysis using the AATI Fragment Analyser; Fluorimetric quantitation using the Invitrogen Qubit). Poly-A mRNA was selected, and MGIEasy stranded mRNA chemistry was performed. All of the samples were quantified and analysed with the Qubit and Bio-analyser, and they all passed the QNA QC standard. Micromon (Monash University, Australia) performed the sequencing on MGITech MGISEQ2000RS hardware (MGISEQ-2000RS High-throughput Sequencing Set). The MGIEasy V2 chemistry set was used to generate the library, with at least 400 m raw reads per lane and paired-end 100b reads for accuracy and processing to compressed fastq files. Files were then delivered digitally.

### 2.4. Data Analysis

Monash Bioinformatics conducted the bioinformatics (Monash University, Australia). The RNAsik pipeline version 1.5.4 [61] and the Mus musculus reference, GRCm38 (Genbank accession GCA 000001635.2), were used to analyse raw fastq files. Using featureCounts [62], reads were quantified, yielding the raw gene count matrix and various quality control metrics. Raw counts were then analysed with Degust, a web tool that performs differential expression analysis using limma voom normalisation [63], producing counts per million library size normalisation and trimmed means of M values and normalisation for RNA composition normalisation [64]. Degust also includes a number of quality plots, such as classic multidimensional scaling and MA plots. To analyse differential gene expression based on overrepresentation analysis, DAVID (The Database for Annotation, Visualisation, and Integrated Discovery) software for KEGG (Kyoto Encyclopedia of Genes and Genomes) pathways was used. EdgeR was preferred to obtain the total number of genes from Degust. An absolute log fold change (abs log fc) of 1 and a false discovery rate (FDR) of 0.05 were selected as the cut-off.

## 3. Results and Discussion

The *Winnie* mouse model, a model of spontaneous chronic colitis, was used to study intestinal alterations, intrinsic and extrinsic innervations of the colon, and the interactions with the enteric nervous system. However, the potential of *Winnie* mice to act as a pre-cancerous model, especially following its progression to severe inflammation which results in rectal prolapse (*Winnie-Prolapse* mice), has not been investigated thus far. By analysing the transcriptome of *Winnie* vs. *Winnie-Prolapse* and comparing it to wild-type C57BL/6 control mice, we may have provided novel insights into the relationship between inflammation and cancer and whether long-standing inflammation has a higher rate of progression into cancer. Herein, the presence of checkpoint markers and cancer markers in *Winnie* and *Winnie-Prolapse* mice were determined and compared to age-matched C57BL/6 mice for analysis (Figure 1). The presence of these markers will give new insights into new targets for immunotherapy in chronic colitis/IBD and their role in the progression to CAC/CRC. The current study compared the gene expression of checkpoint and cancer genes of C57BL/6 control mice, *Winnie* mice, and severely inflamed *Winnie-Prolapse* mice.

### 3.1. Pathway Analysis

#### 3.1.1. Inflammatory and Oncogenic Pathways Are Upregulated in *Winnie* (Colitis—Inflammation) Mice with Further Upregulation in *Winnie-Prolapse* (Severe Colitis—Severe Inflammation) Mice

Inflammation pathways, cell adhesion molecules, neuroactive ligand–receptor interaction pathways, and pathways of viruses are upregulated in *Winnie* mice, with metabolic and cancer pathways significantly expressed in *Winnie-Prolapse* mice. Here, we provide insights into pathways involved in the progression of inflammation to cancer.

Pathway analysis of the transcriptome showed that C57BL/6 and *Winnie* mice co-expressed genes that were significantly enriched in neuroactive ligand–receptor interaction, cytokine–cytokine receptor interaction, and cell adhesion molecules (Figure 2A). C57BL/6 and *Winnie-Prolapse* mice co-expressed genes that were significantly overexpressed in metabolic pathways, pathways in cancer, and neuroactive ligand–receptor interaction (Figure 2B). Both *Winnie* and *Winnie-Prolapse* mice showed overlaps in several pathways, including cell adhesion and cytokine–cytokine receptor interaction (Figure 2A,B). Further, the neuroactive ligand–receptor interaction pathway is among the most highly enriched categories in *Winnie* and *Winnie-Prolapse* mice. Oxidative stress and inflammation, both of which have been shown to be highly present in patients with chronic colitis, were also significantly exhibited by *Winnie* mice [65]. Oxidative stress is a critical factor in the pathogenesis of CRC, which indicates that the functions and roles of neuroactive ligand–receptor interaction in CRC are worthy of further investigation [66].

It is evident that the transcriptome of the distal colon contains a greater number of genes associated with immune- and inflammatory-related pathways, which involve cytokines and chemokines that stimulate cancer cell growth, disrupt differentiation, and aid in cancer cell survival. Thus, it is not only important for biomarker development but also for novel therapeutic targets.

Chronic colitis in *Winnie* mice showed increased metabolite production and alterations to several metabolic pathways, primarily affecting amino acid synthesis and the breakdown of monosaccharides to short-chain fatty acids [67]. This also leads to the fact that the severely inflamed *Winnie-Prolapse* mice were greatly enriched in metabolic pathways. These supply energy and vital metabolites to cancer cells which in turn promote biosynthesis, proliferation, and other important processes of tumourigenesis. Interestingly, pathways of virus infections of human T-cell leukemia virus 1, human papilloma virus, Epstein–Barr virus, Kaposi sarcoma-associated herpesvirus, and human immunodeficiency virus were highly enriched in both *Winnie* and *Winnie-Prolapse* mice (Figure 2A,B), suggesting chronic inflammation in these mice follows the virus inflammation pathways, all of which are known to be involved in the development of cancer.

The pathways in the *Winnie* mice are predominantly those relating to inflammation and virus pathways. Even though these are also present in *Winnie-Prolapse* mice, the most highly expressed pathways are those that are metabolic and pathways involved in cancer (Figure 3), which align with the hypothesis of severe inflammation leading to cancer. The cancer-related pathways which are expressed by both *Winnie* and *Winnie-Prolapse* mice include *Pd-L1/Pd-1* checkpoint pathways, Toll-like receptor pathways, C-type lectin receptor signalling pathways, pro-inflammatory *Il-17/Th17/Th1/Th2/Tnf* signalling pathways, JAK-STAT signalling pathways, and cell adhesion molecules (Figure 3A). Of note, pathways which are only expressed in *Winnie-Prolapse* mice but not in *Winnie* mice are those of chemical carcinogenesis, choline metabolism in cancer, as well as signalling pathways of RAS, Rap1, mitogen-activated protein kinase (MAPK), and PI3K-Akt (Figure 3B). RAS, Rap1m, and JAK-STAT pathways have been shown to be the main drivers in the oncogenic circuitry which support cancer cell growth, migration, survival, invasion, and metastasis [68,69,70]. In addition, the MAPK pathway is involved in the regulation of genes, uncontrolled cell proliferation, cancer-promoting properties, and the resistance of cells to apoptosis [71]. As such, MAPK inhibitors have been used for the management and treatment of cancer. In fact, MAPK is aberrantly activated in patients with CRC, and the inhibition of the MAPK pathway enhances the cytotoxic effects of chemotherapeutic drugs in CRC cells [72]. Further, the PI3K-Akt, TGF-beta, and Wnt signalling pathways are linked to the development of CRC [73]. In patients with IBD, via high-throughput sequencing technology, mutations in RTK/RAS, PI3K, WNT, and TGF-beta pathways have been shown [74]. *Winnie* mice express genes of inflammation and virus pathways, which progress to pathways of cancer in *Winnie-Prolapse* mice. Identifying and understanding genes in cancer pathways and their progression from inflammation to severe inflammation to cancer may translate to improved treatments for cancer patients.

A combination of violin plots, box plots, and jitter plots is shown in Figure 4A regarding differentially expressed genes associated with cancer. The most upregulated gene in *Winnie* mice was *Ifn-γ* (Figure 4B). The overexpression of *IFN-γ* has been linked with the upregulation of *IDO1*, *CTLA-4*, *TIM3*, *PD1*, *PD-L1*, and *LAG3* [75]. As shown herein, these genes are significantly upregulated in *Winnie* mice. In addition, calmodulin-4 (*Calm-4*) (log fc 11.9) was the most upregulated in *Winnie-Prolapse* (Figure 4C). The overexpression of calmodulin-dependent serine kinase (CASK) in conjunction with heparan sulphate proteoglycan syndecan-2 has been linked with the increased metastatic capacity of CRC which affects lymph nodes and increases vascular invasiveness, and liver metastasis [76]. The most downregulated gene in *Winnie* mice was NK3 Homeobox 1 (*Nkx3-1*) (log fc −5.3) (Figure 4D), which has been identified as a prostatic tumour suppressor gene and is downregulated in metastatic prostate cancers [77], but its role in CRC is not clear. The most downregulated gene in *Winnie-Prolapse* mice was shown to be Wnt family member 8B (*Wnt8b*) (log fc −4.0) (Figure 4E). The Wnt signalling pathway is being investigated as a therapeutic target against CRC [78,79].

#### 3.1.2. Upregulation of Key Cellular Adhesion Molecules Which Act as Checkpoint Markers in Winnie Mice

The pathway analysis showed the differential expression of checkpoint markers that are associated with cell adhesion molecules (CAMs), PD-L1 expression, and PD-1 pathway in *Winnie* and *Winnie-Prolapse* mice.

Intercellular and cell–extracellular matrix interactions in cancer involve a variety of cell adhesion molecules [80]. A STRING protein–protein interaction network of CAMs and Pd-L1 expression and the Pd-1 pathway are shown in Figure 5A. The pathway analysis of the transcriptome showed that CAMs were overexpressed in Winnie and Winnie-Prolapse mice when compared with age-matched C57BL/6 control mice (Figure 5A–C). The most upregulated CAM in Winnie as well as Winnie-Prolapse mice was Ctla-4 (log fc 3.8 in Winnie and 4.5 in Winnie-Prolapse) (Figure 5D,E). Ctla-4 is an inhibitory immune checkpoint present on tumour-infiltrating lymphocytes and has been evaluated in several types of cancers. In CRC, Ctla-4 promotes tumour growth and metastasis [81], as well as silencing CRC cell lysate-loaded dendritic cells for immunotherapy [82,83]. The inhibition of Ctla-4 can reactivate T cells in patients with CRC, boosting their immune cells against cancer cells [84].

Highly expressed genes also included histocompatibility 2 (H2) antigens, *H2-Dmb1*, *H2-Dma*, *H2-Q6*, *H2-Q7*, *H2-Ab1*, *H2-Q10*, and *H2-Aa* (Figure 5D,E). T-cell immunoreceptors with Ig and ITIM domains (*Tigit*) (log fc 3.2 in *Winnie* and 3.6 in *Winnie-Prolapse*) and *Pd-L1/Cd274* (log fc 2.2 in *Winnie* and 3.2 in *Winnie-Prolapse*) were upregulated in *Winnie* and *Winnie-Prolapse* mice (Figure 5D,E). Based on previous studies, the upregulation of *TIGIT* and *PD-1* in CRC with mismatch repair deficiency is linked to the tumour node metastasis stage and disease-free survival [85], which makes it a target of interest. The inducible T-cell co-stimulator (*Icos*) which belongs to the B7-CD28 immunoglobulin superfamily was shown to be upregulated in *Winnie* (log fc 1.9) and *Winnie-Prolapse* (log fc 2.7) mice (Figure 5D,E). *Icos* has gained attention due to its expression in several cancers, and its expression in CRC is associated with improved survival, and it may be an appropriate biomarker for CRC [86].

*Selp*/*Cd62* (Selectin P) (log fc 1.8 in *Winnie* and 4.0 in *Winnie-Prolapse*) and *Itga2b* (Integrin alpha 2b) were upregulated in *Winnie* (log fc 1.1) mice and significantly upregulated in *Winnie-Prolapse* (log fc 2.5) mice. *Itga2b* [87] and *Selp* [88] are cancer-related genes with high value in cancer detection. In addition, *Selplg*/*Cd162* (P-selectin glycoprotein ligand-1) and *Itgal* (Integrin Subunit Alpha L) were upregulated in *Winnie* (log fc 1.4 and 1.4, respectively) and significantly overexpressed in *Winnie-Prolapse* mice (log fc 2.5 and 2.5). In CRC, a lack of *Selplg* is linked to CRC growth and may be a good diagnostic or prognostic biomarker [89]. *Itgal* plays a key role in cancer development, the construction of the tumour microenvironment, and in the angiogenesis of cancers including CRC [90].

*Cd6* was upregulated in *Winnie* and highly overexpressed in *Winnie-Prolapse* mice, which is overexpressed in the inflamed mucosa of patients with IBD and accelerates intestinal mucosal immune responses via promoting CD4+ T cell proliferation and differentiation into Th1/Th17 cells [91]. The *Cd6* ligand is also associated with the aggressiveness and metastatic potential of human cancers. Thus, *Cd6* may serve as a novel therapeutic target for the treatment of IBD and cancer immunotherapy in the future [92].

In addition, there were 15 downregulated CAMs (Figure 5D), of which, the most downregulated CAMs in *Winnie* (log fc −3.0) and *Winnie-Prolapse* (log fc −4.0) mice were histocompatibility 2, T region locus 3 (*H2-T3*), which is mainly involved in protein binding. Cadherin 3 or P-Cadherin (*Cdh3*) were downregulated in *Winnie* (log fc −1.3) but upregulated in *Winnie-Prolapse* (log fc 1.0) mice. It is abundant in the placenta and is overexpressed in several cancers, but the expression levels in CRC are not clear [93].

The roles of *Pd-1*, *Pd-L1*, *Ctla-4*, *Ido1*, *Tim3,* and *Lag3* (Figure 6) and their upregulation have been studied extensively in various types of cancers, including CRC in humans. These markers were shown to be upregulated in both *Winnie* and *Winnie-Prolapse* mice compared to C57BL/6 mice. In addition, high expression levels of *Cd45* (also known as *Ptprc*) (Figure 6) was noted in *Winnie* and *Winnie-Prolapse* mice. Increased *Cd45* expression in CRC cells in pre-treated primary cancers is responsible for poor regression and the recurrence-free survival of patients [94]. Further, primary tumour volume and secondary metastases are reduced by targeting *Cd276* [95]. The expression of *Cd276* was also noted to be upregulated in *Winnie* and *Winnie-Prolapse* mice compared to the controls. Moreover, *Cd177* was previously shown to be highly expressed in CRC/CAC [96] and was also shown to be highly upregulated in both *Winnie* (log fc 3.1) and *Winnie-Prolapse* (log fc 3.5) mice; it is certainly a target of interest in the treatment and prognosis of inflammation, IBD, and cancer.

#### 3.1.3. Transcriptomes of *Winnie-Prolapse* Mice Are Enriched in Cancer-Related Pathways That Are Absent in *Winnie* Mice

As noted in Figure 3, the pathway analysis of *Winnie-Prolapse* and C57BL/6 mice showed great enrichment in cancer-related pathways, especially the PI3K-AKT signalling pathway, MAPK signalling pathway, RAP1 signalling pathway, RAS signalling pathway, and NF-kB signalling pathway. In particular, the RAS protein operates in two main cellular pathways—the mitogen-activated protein kinase (MAPK) and phosphoinositide-3 kinase (PI3K) pathways [97]. These are vital in controlling various processes in a normal cell, including cell growth and survival. The activation of the PI3K-Akt signalling pathway has been noted in 60–70% of patients with CRC [98], and as such, inhibitors of route components could be used as therapeutic drugs [99]. The PI3K-AKT signalling pathway regulates the cell cycle and is involved in the quiescence, proliferation, and development of cancer. Within the PI3-AKT pathway, genes which are highly expressed in *Winnie-Prolapse* mice include *Ccnd1*, *Col1a1*, *Hsp90b1*, *Itga6*, and *Lamc2* (Figure 7A). Genes involved in Ras signalling are shown in Figure 7B, and those involved in the NF-kB signalling pathway are depicted in Figure 7C.

The Ras/Raf/MEK/ERK cascade is involved in the control of growth signals, cell survival, and invasion in cancer [100]. Proteoglycans in cancer are situated in the glycocalyx of cancer cells; proteoglycans provide a contact link between the cell membrane and the surrounding extracellular matrix, thereby playing a central role in regulating cancer cell adhesion and migration [101]. The nuclear factor-kappa B (NF-kB) signalling pathway has been linked to carcinogenesis as a regulator of immune response and inflammation [102]. NF-kB may contribute to the progression of CRC [103]. The therapeutic stimulation of cGMP/PKG pathways provides a promising avenue for CRC [104] prevention and treatment; however, further preclinical data are required to fully understand their role. Rap1 signalling regulates integrin- or cadherin-mediated cell adhesion, protease levels (e.g., matrix metalloproteinase), and cytoskeletal alterations, all of which are linked to cancer cell proliferation, invasion, and metastasis [70]. Active Rap1 inhibits tumour invasion in several cancers, implying that Rap1 signalling has cancer-specific pleiotropic effects [105]. Targeting Rap1 signalling and its controllers may control carcinogenesis, metastasis, chemoresistance, and immunological evasion [106].

### 3.2. Gene Analysis

#### Transcriptomes of *Winnie* and *Winnie-Prolapse* Mice Are Highly Expressed in Cancer Genes (Oncogenes)

Genes from C57BL/6, *Winnie*, and *Winnie-Prolapse* mouse colon tissues were analysed on The International Cancer Genome Consortium Data Portal (ICGC Data Portal) (http://www.icgc.org/ URL accessed on 21 February 2023). There were 115 genes from our data that matched the cancer gene database. Out of these, the DEGs from C57BL/6 and *Winnie* mice showed 50 (32 upregulated and 18 downregulated) similar genes (Figure 8A), and 106 DEGs (75 upregulated and 31 downregulated) were found to be similar in C57BL/6 and *Winnie-Prolapse* mice (Figure 8B). The lists of the top 10 up- and downregulated cancer genes in *Winnie* and *Winnie-Prolapse* mice are shown in Figure 8C–F.

Colony-stimulating factor 3 receptor (*Csf3r*) was the most upregulated gene in *Winnie* (log fc 3.2) and *Winnie-Prolapse* mice (log fc 5.4). (Figure 8C,D). *CSF3* and *CSF3R* are highly expressed in CRC, which are important regulators which promote pro-tumour behaviour in cancer and immune cells in CRC. *Csf3r* correlates with many genes linked to poor prognosis in CRC [107]. Upon analysing the correlations of *CSF3* and *CSF3R* expression and various factors such as patient demographics, tumour stage, and consensus molecular subtype using data from The Cancer Genome Atlas Firehose Legacy, it was noted that the levels of *CSF3* and *CSF3R* expression were most elevated in patients belonging to consensus molecular subtype-1 and consensus molecular subtype 4. In addition, the intestine-specific homeobox (*Isx*) (log fc −3.5) (Figure 8E) gene was found to be highly downregulated in *Winnie* mice, which aligns with earlier research findings that indicate *Isx* experiences downregulation in colon adenomas [108]. Insulin-like growth factor 2 mRNA-binding protein 2 (*Igf2bp2*) was the most downregulated in *Winnie-Prolapse* mice (log fc −3.4) (Figure 8F) and is known to promote CRC progression in humans [109], and its dysregulation is associated with the progression of cancers and cancer stem cells [110]. Several cancer genes show significant differential expression between *Winnie* and *Winnie-Prolapse* mice (Figure 8G), including Myeloid leukemia factor 1 (*Mlf1*) (log fc 0.06 in *Winnie* and 3.9 in *Winnie-Prolapse*) and GATA binding protein 3 (*Gata3*) (log fc 0.6 in *Winnie* and 3.9 in *Winnie-Prolapse*), but their roles and functions in CRC are unclear.

### 3.3. Other Genes Linked to CAC and CRC

Other genes overexpressed in the transcriptome of *Winnie* (log fc 7.7) and *Winnie-Prolapse* (log fc 8.7) mice include chloride channel accessory 4 (*CLCA4*) a human ortholog of *Clca4b*; its overexpression and function is to inhibit migration and invasion by suppressing the epithelial-to-mesenchymal transition via PI3K/ATK signalling, and it predicts the favourable prognosis of CRC [111]. Previous studies have shown improved overall survival in patients with various types of cancer, including CRC, by decreasing the expression of *CLCA4* [112]. In fact, it was noted that the downregulation of CLCA4 expression correlates to the progression of CRC, breast cancer, stomach cancer, and head and neck cancers [112]. Further, CLCA4 is significantly reduced in oesophageal cancer tissues, and its expression correlates with stage, differentiation, lymph node involvement, and metastatic disease [113]. Transporter associated with antigen processing 1 (*Tap1*) was significantly overexpressed in both *Winnie* (log fc 2.7) and *Winnie-Prolapse* (log fc 3.3) mice, associated with antigen processing and presentation pathways. Reduced *Tap1* expression is linked to poor prognosis in patients with CRC and is linked to immune checkpoint genes [114], DNA methylation, tumour mutation burden, microsatellite instability, and neoantigens. *Tap1* could be a potential therapeutic target as well as a biomarker for the early detection of cancer [114,115]. The study in this paper confirms our previous findings that 1% dextran sulphate sodium administration in *Winnie* mice induces colonic changes displaying dysplasia in 55% of mice. The expression levels of *Cav1* and *Trp53* were shown to be regulated differently in the distal colon of *Winnie* mice compared to control mice. Additionally, the expression of *Myc* and *Ccl5* was specifically elevated in *Winnie* mice, and *Ccl5* was associated with greater complexity in abnormal crypts [47]. In humans, the main genetic alterations in IBD-CRC compared to sporadic CRC are *p53* and *TGFBR2* [74], which were also expressed in *Winnie* and *Winnie-prolapse* mice.

## 4. Future Prospects

The regulation of the tumour microenvironment based on multiple omics results can suggest innovative therapeutic strategies to prevent cancer cells from immune escape and to support anti-cancer effects. Bioinformatics approaches will also need to continue to develop to improve our understanding of a responsive tumour microenvironment. It is well established that the tumour microenvironment affects immune checkpoint inhibitor responses. Predictive and prognostic biomarkers can be determined using cutting-edge technology, which can also be used to determine the immunological background of malignancies. Though there has been significant research and information on checkpoint markers, their roles in IBD and the development of CRC/inflammation-associated cancer remains unclear. Studying checkpoint markers in the *Winnie* mouse models that closely resemble human models may lead to the discovery of biomarkers for screening IBD patients and for the advancement of understanding inflammation and cancer, which may lead to designing drugs that use inhibitors of biomarker expression or that can be used in vaccines to prevent disease progression. This includes patients at high risk of disease relapse who may benefit from earlier treatments. Although the clinical development of immune checkpoint inhibitor therapy has ushered in a new era of anti-cancer therapy, with sustained responses and significant survival advantages observed in multiple cancers, most patients do not benefit.

## 5. Conclusions

The current study has revealed the notable overexpression of cell adhesion molecules and associated pathways in cancer, particularly in *Winnie-Prolapse* mice when compared to *Winnie* mice. The elevated presence of oncogenes and pathways linked to cancer in *Winnie-Prolapse* mice as compared to *Winnie* mice suggests a potential connection between inflammation and cancer, even in pre-cancerous stages. This leads to the hypothesis that chronic and progressive inflammation may contribute to the development of cancer. Despite the recent focus on checkpoint markers in the last five years, their specific functions in the context of IBD and the inflammation that leads to CRC remain unclear and warrant further investigation. Given the similarities observed between the transcriptome of human IBD patients and spontaneous IBD models like *Winnie* mice, as well as the severely inflamed *Winnie-Prolapse* model, it is valuable to conduct more in-depth studies on these checkpoint markers. This research can help elucidate their roles in inflammation and their potential relationship with the progression to cancer.

The advent of immune checkpoint inhibitor therapy has revolutionised cancer treatment, leading to sustained responses and significant survival benefits in various cancers. Despite these promising outcomes, many patients do not experience these advantages. Consequently, there is interest in identifying predictive biomarkers, from early inflammation to severe inflammation, that can indicate the development of cancer. Combining chemotherapy with checkpoint inhibitor immunotherapies is becoming increasingly common. While immune checkpoint blockade has demonstrated the ability to generate lasting responses in several cancer types, these remarkable outcomes are currently observed in only a minority of patients and specific indications. As such, there is an urgent need for more effective and innovative approaches.

In summary, this study identified genes associated with chronic colitis (inflammation) and the progression to prolapse (severe inflammation), particularly in *Winnie-Prolapse* mice, revealing several genes enriched in cancer pathways.

## Figures and Tables

**Figure 1 cancers-15-04793-f001:**
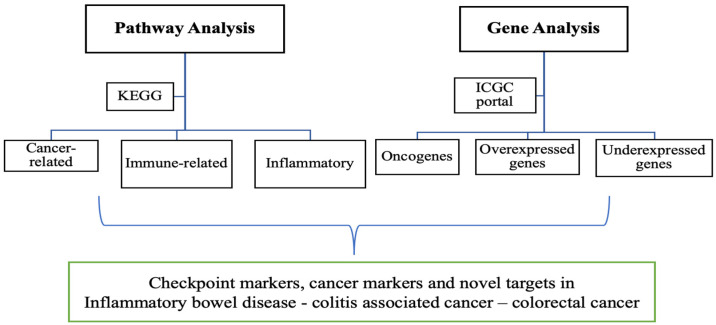
Flowchart—Overview of pathway and gene analysis.

**Figure 2 cancers-15-04793-f002:**
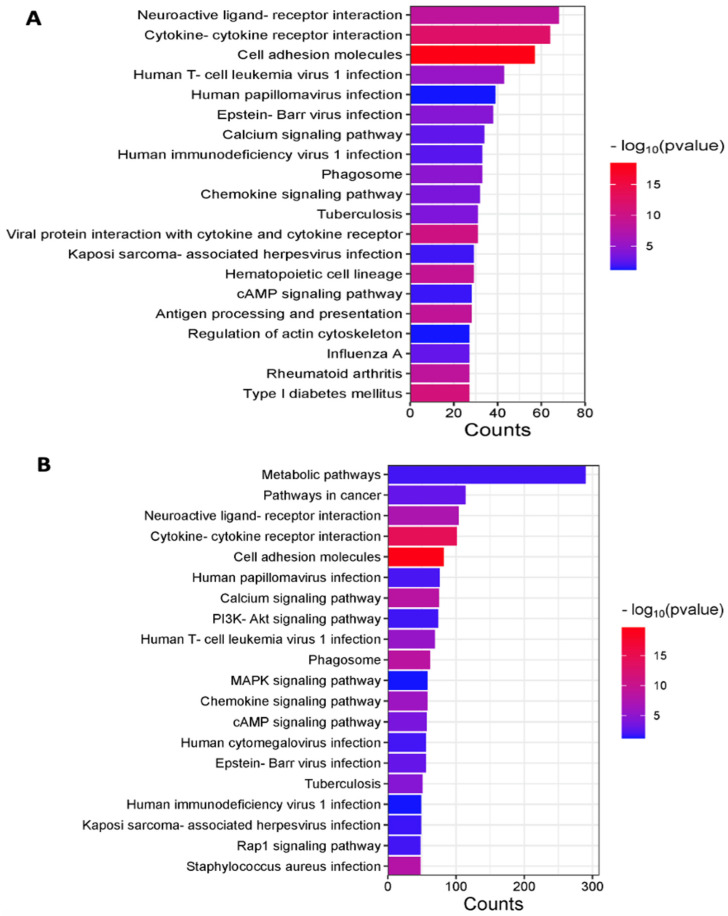
Pathway Analysis—KEGG bar chart, where the number of genes is shown on the x-axis and the KEGG pathways are represented on the y-axis. (**A**) Top 20 KEGG pathways from genes co-expressed by C57BL/6 and *Winnie* mice and (**B**) top 20 KEGG pathways from genes co-expressed by C57BL/6 and *Winnie-Prolapse* mice. The length of the bar represents the number of genes, and the colour represents the significance of the log_10_ *p* value.

**Figure 3 cancers-15-04793-f003:**
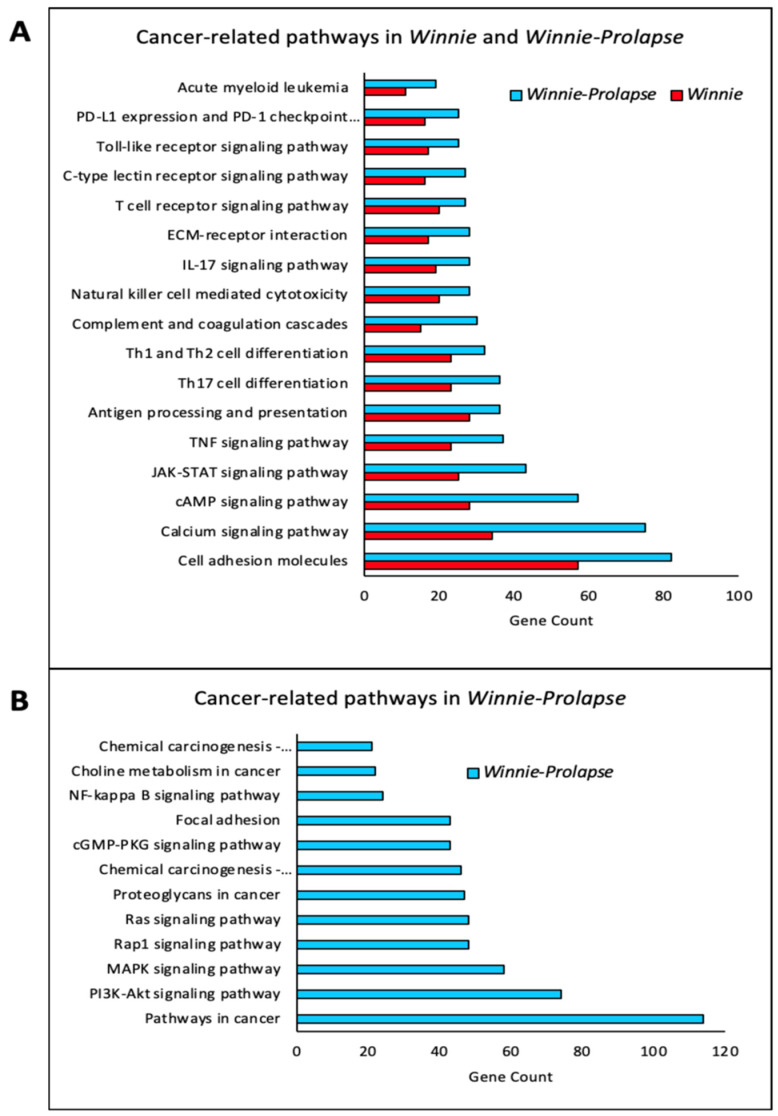
Bar chart where the number of genes is shown on the x-axis and the KEGG pathways are represented on the y-axis. (**A**) Cancer-related KEGG pathways in *Winnie* and *Winnie-Prolapse* mice. (**B**) Cancer-related pathways exhibited only in *Winnie-Prolapse* and absent in *Winnie* mice. The length of the bar represents the number of genes.

**Figure 4 cancers-15-04793-f004:**
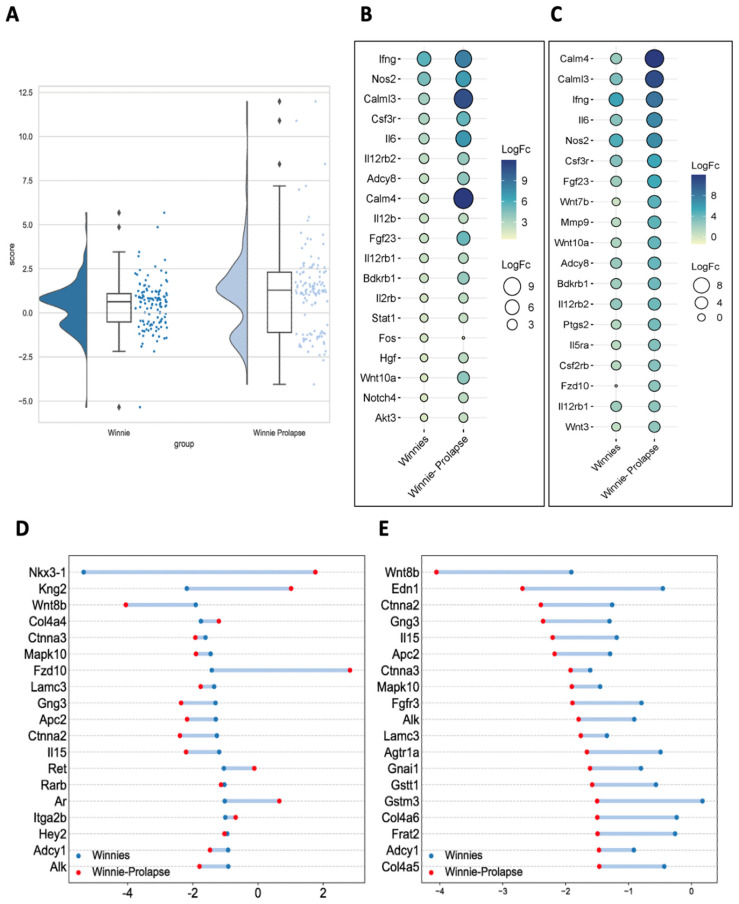
Gene expression (DEGs in pathways in cancer)—Raincloud plot (combination of violin plot, box plot, and jitter plot). (**A**) Number of differentially expressed genes associated with pathways in cancer in *Winnie* mice. (**B**) Top 20 upregulated genes associated with pathways in cancer. (**C**) Top 20 upregulated genes associated with pathways in cancer in *Winnie-Prolapse* mice. (**D**) Top 20 downregulated genes associated with pathways in cancer in *Winnie* mice. (**E**) Top 20 downregulated genes associated with pathways in cancer in *Winnie-Prolapse* mice.

**Figure 5 cancers-15-04793-f005:**
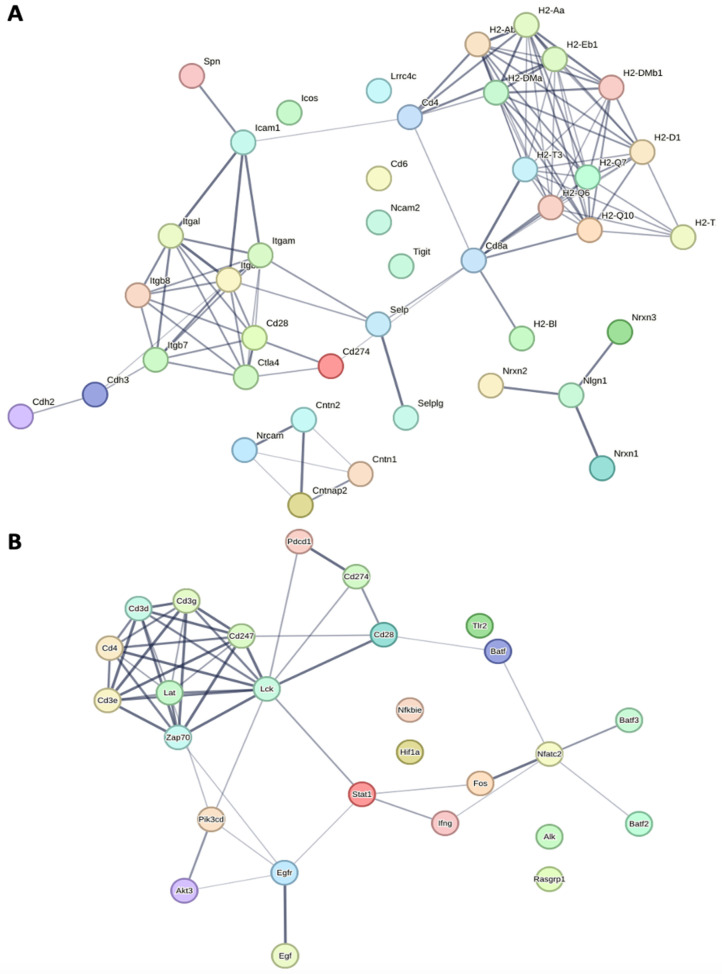
Gene expression (CAMs). (**A**,**B**) STRING network—each circle represents one gene, and the length of the node represents the level of interaction. (**A**) Protein–protein interaction between genes associated with cell adhesion molecules and (**B**) protein–protein interaction between genes associated with PD-L1 expression and PD-1 checkpoint pathways. (**C**) Heatmap of differential expression of genes associated with cell adhesion molecules across C57BL/6, *Winnie* and *Winnie-Prolapse* mice based on the Z score, hierarchically clustered based on Euclidean distance. Top 20 upregulated genes associated with cell adhesion molecules in (**D**) *Winnie* and (**E**) *Winnie-Prolapse* mice. (**F**) Downregulation of genes associated with cell adhesion molecules in *Winnie* and *Winnie-Prolapse* mice. C57BL/6 mice (C1–C8), *Winnie* mice (W1–W7), and *Winnie-Prolapse* mice (WP1–7).

**Figure 6 cancers-15-04793-f006:**
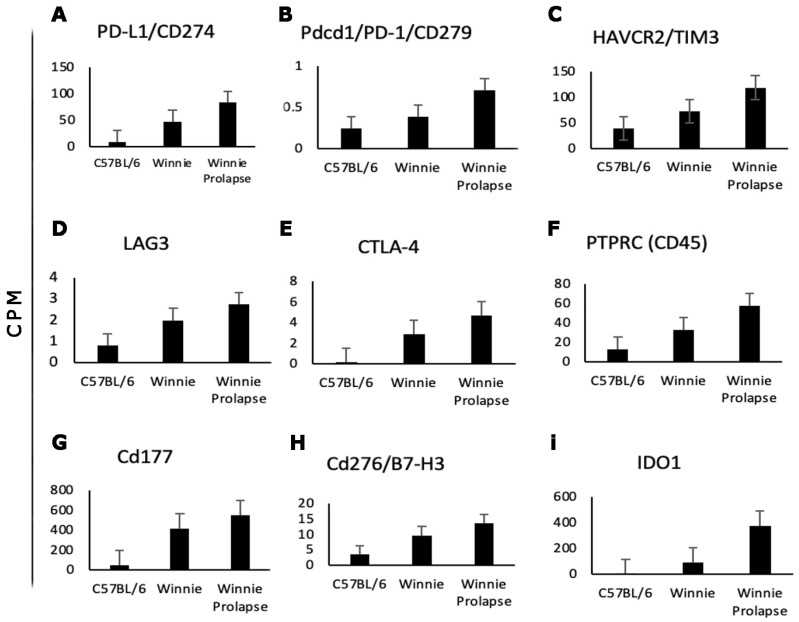
Expression of checkpoint markers. Bar graphs showing differential gene expression in counts per million (CPM; y-axis) for checkpoint markers. Expression is shown for C57BL/6, *Winnie,* and *Winnie-Prolapse* mice where checkpoint markers are enhanced in *Winnie* and even further enhanced in *Winnie-Prolapse* mice. Bars represent mean ± standard deviation.

**Figure 7 cancers-15-04793-f007:**
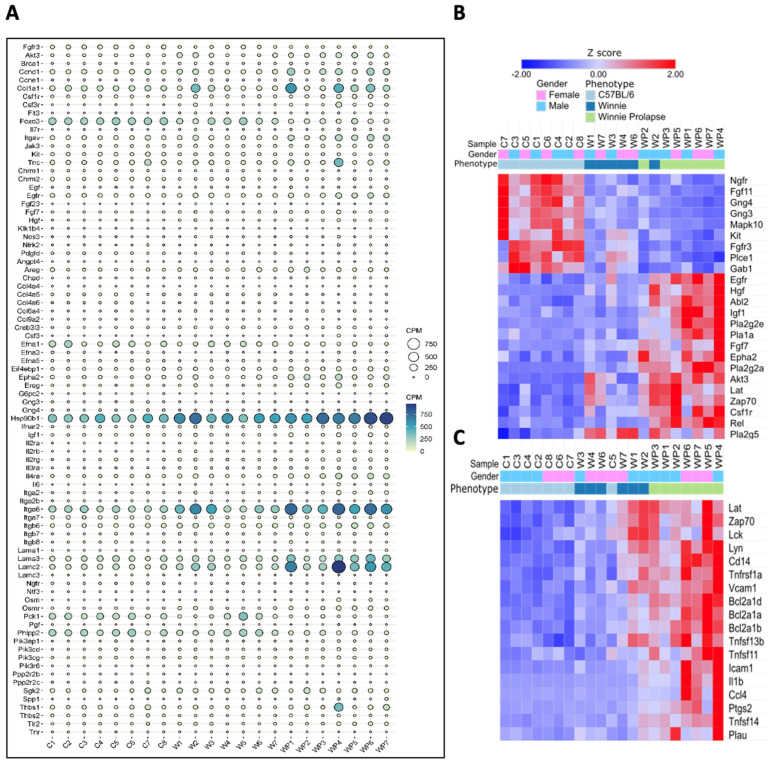
Gene expression. (**A**) Differential gene expression associated with the PI3-AKT signalling pathway; the size of the bubble represents counts per million (CPM). (**B**) Heatmap of genes involved in the Ras signalling pathway and (**C**) heatmap of genes involved in the NF-kB signalling pathway. C57BL/6 mice (C1–C8), *Winnie* mice (W1–W7), and *Winnie-Prolapse* mice (WP1–7).

**Figure 8 cancers-15-04793-f008:**
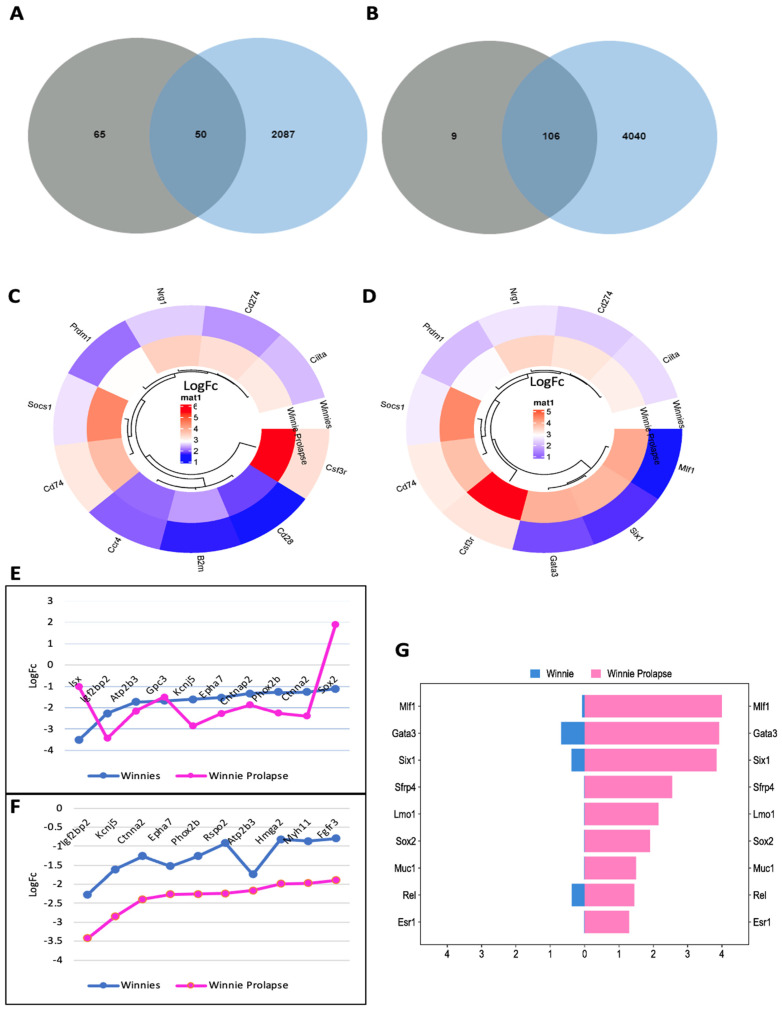
Cancer gene analysis. Venn diagram of similar genes between the cancer genes from the ICGC portal (cancer gene consortium) and (**A**) C57BL/6/*Winnie* mice or (**B**) C57BL/6/*Winnie-Prolapse* mice. (**C**) Top 10 upregulated cancer genes in *Winnie* mice, (**D**) top 10 upregulated cancer genes in *Winnie-Prolapse* mice, (**E**) top 10 downregulated cancer genes in *Winnie* mice, and (**F**) top 10 downregulated genes in *Winnie-Prolapse* mice. (**G**) Significant gene expression in *Winnie-Prolapse* when compared to *Winnie* mice.

## Data Availability

Data available from the authors on request.

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
