# Peer review of "Differential Gene Expression of Checkpoint Markers and Cancer Markers in Mouse Models of Spontaneous Chronic Colitis"

_cancers, 2023, doi:10.3390/cancers15194793_

Round 1

Reviewer 1 Report

I thank the authors for submitting the manuscript to the journal. The methods followed adequately support the conclusions of the manuscript.

Author Response

Comment: I thank the authors for submitting the manuscript to the journal. The methods followed adequately support the conclusions of the manuscript.

A: Thank you

Reviewer 2 Report

Reviewer comments and suggestions

The authors in this study highlighted the importance of checkpoint makers, cancer-related pathways, and cancer genes in colon tissues of mouse models of chronic colitis (Winnie and Winnie-Prolapse mice) using next-generation sequencing. Identification of such genes and their subsequent expression and role at the protein level would enable novel markers for the early diagnosis of cancer in IBD patients. The study reported that several oncogenes is associated with the severity of IBD, with Winnie-Prolapse mice expressing a large number of key genes associated with the development of cancer. 

Overall, the manuscript was well written. However, a few major concerns/comments needed to be explained/modified. 

  1. Line 46-48 Need references
  2. Reference 13-15 it would be nice if the authors could cite research articles rather than review the manuscript
  3. Line 57-62 this line should be shorten as the information as similar
  4. Line 78-79 is there was a report to this drug, please mention the reference and explain accordingly
  5. Line 231-232 What are the differences between these two?
  6. Line 246-247 I have seen the explanation of both mice in Figure 2 and Figure 3, it seems that the authors discuss it more,
  7. For figure 5 A. is this important that a few points were not interacted?
  8. Line 451-452 Please mention the references
  9. Line 458 Please check the cited references [105,107]
  10. Section 4 I could not find any limitations in the paragraphs
  11. Comments for conclusion paragraph “In my thinking, the conclusion should be short and provide a novel statement that worked out from this study, no need to add references to support this data”.
  12. All references should be modified based on MDPI

Author Response

The authors in this study highlighted the importance of checkpoint makers, cancer-related pathways, and cancer genes in colon tissues of mouse models of chronic colitis (Winnie and Winnie-Prolapse mice) using next-generation sequencing. Identification of such genes and their subsequent expression and role at the protein level would enable novel markers for the early diagnosis of cancer in IBD patients. The study reported that several oncogenes is associated with the severity of IBD, with Winnie-Prolapse mice expressing a large number of key genes associated with the development of cancer. Overall, the manuscript was well written. However, a few major concerns/comments needed to be explained/modified. 

  1. Line 46-48 Need references.

A: Thank you, added

  1. Reference 13-15 it would be nice if the authors could cite research articles rather than review the manuscript

A: Thank you, added

  1. Line 57-62 this line should be shorten as the information as similar

A: Thank you, edited accordingly

  1. Line 78-79 is there was a report to this drug, please mention the reference and explain accordingly.

A: included

  1. Line 231-232 What are the differences between these two?

A: Winnie and Winnie prolapse mice, their differences are made clear in the methods section, Thank you

  1. Line 246-247 I have seen the explanation of both mice in Figure 2 and Figure 3, it seems that the authors discuss it more,

A: the pathways to cancer are all mentioned in lines 227-250

  1. For figure 5 A. is this important that a few points were not interacted?

A: they are all different pathways, so a few in 5A that are not interacted suggests they are separate pathways which do not overlap.

  1. Line 451-452 Please mention the references

A: Thank you, added references and extra comment

  1. Line 458 Please check the cited references [105,107]

A: Checked and fixed, thank you

  1. Section 4 I could not find any limitations in the paragraphs

A: Our error, I have edited this to future prospects

  1. Comments for conclusion paragraph “In my thinking, the conclusion should be short and provide a novel statement that worked out from this study, no need to add references to support this data”.

A: references removed and shortened, as per suggestion

  1. All references should be modified based on MDPI

A: modified to MDPI style

Reviewer 3 Report

The publication: "Differential gene expression of checkpoint markers and cancer markers in mouse models of spontaneous chronic colitis" concerns a very interesting and clinically important problem.

The publication presented the expression of several oncogenes which are associated with the severity of spontaneous chronic colitis and a number of new targets to evaluate for development of biomarkers and therapeutics. The publication indicates top 20 upregulated/downregulated genes associated with pathways in cancer/pathways in cancer in Winnie-Prolapse mice/pathways in cancer in Winnie mice.

I believe that the publication is very well written, uses extensive bioinformatics analyzes and presents the obtained results in a clear way.

I have two comments that may contribute to an even better reception of the publication by readers:

1. In the introduction, describe in more detail the planned purpose of the work.

2. It is interesting whether the indicated genes / selected genes in the mouse model coincide with the results of clinical trials / on human material, can the authors add such information in the discussion?

Congratulations on an interesting publication.

Author Response

Comment: The publication: "Differential gene expression of checkpoint markers and cancer markers in mouse models of spontaneous chronic colitis" concerns a very interesting and clinically important problem. The publication presented the expression of several oncogenes which are associated with the severity of spontaneous chronic colitis and a number of new targets to evaluate for development of biomarkers and therapeutics. The publication indicates top 20 upregulated/ downregulated genes associated with pathways in cancer/pathways in cancer in Winnie-Prolapse mice/pathways in cancer in Winnie mice. I believe that the publication is very well written, uses extensive bioinformatics analyses and presents the obtained results in a clear way.

A: Thank you

I have two comments that may contribute to an even better reception of the publication by readers:

  1. In the introduction, describe in more detail the planned purpose of the work.

A: Thank you, we have made this clearer in the introduction

  1. It is interesting whether the indicated genes / selected genes in the mouse model coincide with the results of clinical trials / on human material, can the authors add such information in the discussion?

A: Thank you, this is now included

Congratulations on an interesting publication

A: Thank you